# Transformation of geological sciences and geological engineering field methods course to remote delivery using manual, virtual, and blended tools in fall 2020

Jennifer Jane Day[1]

[1]Department of Geological Sciences and Geological Engineering, Queen's University, Kingston, K7L 3N6, Canada

*Correspondence to*: Jennifer Jane Day (day.jennifer@queensu.ca)

**Full Institutional Address:**

Geological Sciences and Geological Engineering
36 Union Street
Miller Hall, Queen's University
Kingston, Ontario
K7L 3N6
Canada

**Abstract.** Geological (Engineering) Field Methods (GEOE/L 221) is a core course for two programs at Queen's University in Kingston, Ontario, Canada where students learn foundational knowledge, skills, and methods to conduct field work that is used to investigate geological and geological engineering aspects of the Earth. Typically, this fall-term course involves weekly
field trips in the Kingston area to visit a variety of rock outcrops to learn and practice methods of field navigation, observation, and measurement. Remote delivery of this course in fall 2020 due to COVID-19 without in-person field trips required a significant transformation, which included creating field and demonstration instructional videos, using 3D digital photogrammetry models of rock samples and outcrops, developing independent outdoor activities for pace and compass navigation, manual sketching, and graphical measurements on paper, and utilizing a culminating immersive 3D video game
style geological field mapping exercise. This paper examines these new course elements, how well the course learning objectives were achieved in a remote setting, and the successes and limitations of remote delivery. Although many new virtual elements enhance the course and some have been incorporated into subsequent in-person offerings, a return to in-person teaching for geological sciences and geological engineering field methods courses is strongly recommended.

## 1. Introduction

Geological Field Methods (GEOL 221) and Geological Engineering Field Methods (GEOE 221) are second year core courses in the Geological Sciences and Geological Engineering undergraduate programs, respectively, at Queen's University in Kingston, Ontario, Canada. They are integrated into a single fall term course (GEOE/L 221) where all students participate as

one class and learn the same material, which promotes interdisciplinary learning. The university calendar description of these courses is as follows: *"[the] (engineering) field study of surficial deposits, rock types, and geological processes, based on the*

35 *geology of the Kingston area. Descriptions, samples, and measurements acquired on several field trips will be analyzed, and the results recorded in maps, sections, and reports throughout the course"* (Queen's University, 2020a, 2020b). The author instructed the course in fall 2019 (in person), fall 2020 (remote), and fall 2021 (in person) and was responsible for the course redesign and implementation for remote delivery in fall 2020.

This is primarily a skills-focussed course delivered through lectures, tutorials, field trips, and labs, using the geology of the Kingston area as topical context. Bedrock geology in Kingston features nearly flat-lying early Paleozoic limestones and sandstones that border Precambrian lithologies of the Frontenac Arch (Helmstaedt and Godin, 2008; Carr et al., 2000). The skills learned through this course include field orienteering and navigation, field observation and identification of lithological units, geological structures, and historical geology, measurement of orientations and characteristics of geological structures,

recording of field data and sketches in notebooks and traverse maps, data analysis and synthesis of geological models and creation of geological maps, sections, and stratigraphic columns, structural data analysis using stereonets, use of engineering geology tools to characterize rockmass strength, and professional, integrated reporting of geological and engineering data and interpretations.

This course is also a crucial opportunity for the students to develop their class community that they will interact with in core courses for the rest of their undergraduate degree programs. In the past five years, the course enrollment has consisted of approximately 20-40 students from each of the GEOE and GEOL programs, for a total ranging from approximately 40-80 students. In fall 2020, 24 GEOE and 33 GEOL students were enrolled, for a total class of 57 students.

Geological (engineering) field courses like GEOE/L 221 are essential for students to develop knowledge, skills, and experience of site investigation, which is the source of data for many geoscience and geological engineering projects. Learning in an outdoor and sometimes unfamiliar field environment requires additional preparation including safety training, weather and climate awareness, and making plans to bring appropriate clothing and sustenance. This preparation is an important part of field courses and provides a safe environment for students to learn these skills before they may encounter them in their careers.

Even if students do not participate in field work through their careers, their field experience through courses during their degree programs instills an important respect for field data collection opportunities and challenges, and provides them with practical insight for planning field campaigns and analyzing data collected by others.

In fall 2020, this course was offered remotely on an emergency basis due to the impacts of COVID-19, which required a

65 significant transformation on short notice to deliver the course without field trips or in person tutorials and other labs. The regular in-person activities of live lectures, interactive tutorials, and hands-on field trips and lab periods were redesigned to be

pre-recorded lectures, live virtual tutorials and labs via video conference calls, and pre-recorded demonstration videos about field sites, field skills, and lab skills. These course activities were supplemented with digital tools including online group editing software for interactive written discussions of lectures, readings, and other course content, 3D photogrammetry models

of hand samples and outcrops hosted on Sketchfab (2021), and the Lighthouse Bay immersive virtual field exercise by Houghton and Robinson (2017). Critical hands-on (manual) skill elements were maintained in the remote course delivery, including use of compasses for orienteering and navigation, measurement of geological structures, use of field notebooks for handwritten and drawn field observations and sketches, and use of drafting tools for manually drafted geological maps and sections, stereonets, and topographic contour problems.

This paper describes the Course Learning Outcomes (CLOs), course structure, virtual and manual skills-based learning elements, and community building elements of GEOE/L 221. An analysis of student performance between the in person and remote course offerings is presented, and the successes and limitations of the remote delivery of GEOE/L 221 in fall 2020 are discussed.

**2.   Course Learning Outcomes**

This course and others at Queen's University follow an outcomes-based education model and framework that promotes a learner-centered approach and clarifies competencies of courses and, more broadly, degree programs (McCombs and Whistler, 1997; Weimer, 2002; Pillay, 2002; Kolomitro and Gee, 2015). CLOs are developed at the course level and are mapped to degree program level Graduate Attributes, as part of the Government of Ontario and Canadian Engineering Accreditation

Board program structure requirements (e.g. Hutchinson, 2001; Remenda, 2010). The Geological Sciences and Geological Engineering program curricula at Queen's University have been developed using a Concept Map approach that identifies and maps knowledge and skills into categories of observation and measurement, analysis, design of geological models, and design of engineered solutions involving site investigation programs, monitoring systems, and analysis protocols; all of these categories are linked through development of foundational skills such as ethics, professionalism, communication, judgement,

and teamwork (Remenda, 2010; M. Diederichs, pers. comm.). The CLOs for GEOE/L 221 that were used in fall 2019 (in person), fall 2020 (remote), and fall 2021 (in person) are listed in Table 1.

Table 1: Course Learning Outcomes (CLOs) for GEOE/L 221 Geological (Engineering) Field Methods in fall 2019, 2020, and 2021

| CLO No. | Description |
|---------|-------------|
| CLO-1 | Demonstrate that they can plan and conduct field investigations in a safe, ethical, socially, and environmentally responsible manner with scientific and academic integrity. |
| CLO-2 | Demonstrate facility with basic field and lab techniques for reliable and meaningful measuring and characterizing of key geological and geological engineering parameters. |

| CLO-3 | Categorize and compare the rocks in an area and be able to explain the variability of the characteristics of components in a natural system. |
|---|---|
| CLO-4 | Demonstrate proficiency with basic principles of historical geology which they will be able to use to logically determine the sequence of geological events in an area. |
| CLO-5 | Apply knowledge to solve geological and geological engineering problems with an incomplete or sparse data set in three dimensions. |
| CLO-6 | Begin demonstrating spatial and temporal reasoning on all scales in real time during field work and during analysis of field data. |
| CLO-7 | Select, analyze, synthesize, discuss (oral), and professionally report (written, visual) on geological data as presented on maps and cross-sections. |
| CLO-8 | In groups and individually, critically evaluate geological data and related information from a variety of sources on specific topics in field geology, and report the results in a variety of formats. |
| CLO-9 | Collect and interpret data obtained while on the field trips, and design and submit a written report with maps and recommendations on a site-specific engineering problem. |

**3.    Regular In-Person Course Delivery**

The regular in-person course delivery of GEOE/L 221 is scheduled over a 12-week term. The first 7 weeks of term include weekly field trips in the Kingston area during 4-hour afternoon lab timetable slots. The remaining 5 weeks of term have indoor labs where the students transition from focussing on field observations, data collection, and preliminary analyses, to more advanced data analysis and synthesis of geological models and engineering solutions. The themes of the 7 field trips are listed

in Table 2.

**Table 2: In person field trips for GEOE/L 221 in fall 2019**

| Field Trip No. | Description |
|---|---|
| 1 (On Campus) | Initial learning of field skills including pace and compass navigation, and orientation measurements of planar and linear structural features |
| 2 (Barriefield, Joyceville) | Lithological identification, structural orientation measurements, and age relationships of gently folded Ordovician limestone and jointed Proterozoic syenite outcrops with contacts to intrusive dykes and other younger/older units (Figure 1a) |
| 3 (Moreland-Dixon Road Part 1) | Scanline mapping of outcrop with Proterozoic quartzite, gneiss, mafic dykes, and faults |
| 4 (Perth Road) | Outcrop stops through Proterozoic syenite pluton, including transition from metamorphic country rock and into core of pluton |
| 5 (Wollastonite Mine) | Off-road mapping of folded strata, tour of local Wollastonite mining operation, and engineering geology assessment of rock slope stability |
| 6 (Moreland-Dixon Road Part 2) | Mapping a stratigraphic section through Ordovician limestones (Figure 1c) |
| 7 (Field Exam) | Field exam on rock identification, relative ages of units, and structural measurements |

Each field trip includes a deliverable such as submission of field notebooks, traverse maps, completed geological maps and sections, engineering geology analysis, and stratigraphic sections, all accompanied by an interpretive memo or report. Developing skills in field data collection is emphasized such that students learn how to observe and record geological descriptions, structural measurements, and outcrop sketches, among other data types.

The lectures include topics on geological processes, geological materials, relative ages of rock units, geological models and reporting, engineering geology, economic geology, and a variety of guest lectures from faculty and graduate students in the department to introduce students to the various applications of geological (engineering) field methods. Tutorials are practical and hands-on where students are led through examples of identifying lithologies of hand samples, developing geological models, creating geological maps and cross sections from field data, and analyzing structural data using stereonets. A summative group project uses field data collected by students around the Kingston area to create their own geological maps and sections and report on their interpretation of the geology and geological history of the Kingston area. Examinations included both oral and written formats, where the oral format focused on skills and the written format focused on geological and geological engineering principles and problem solving.

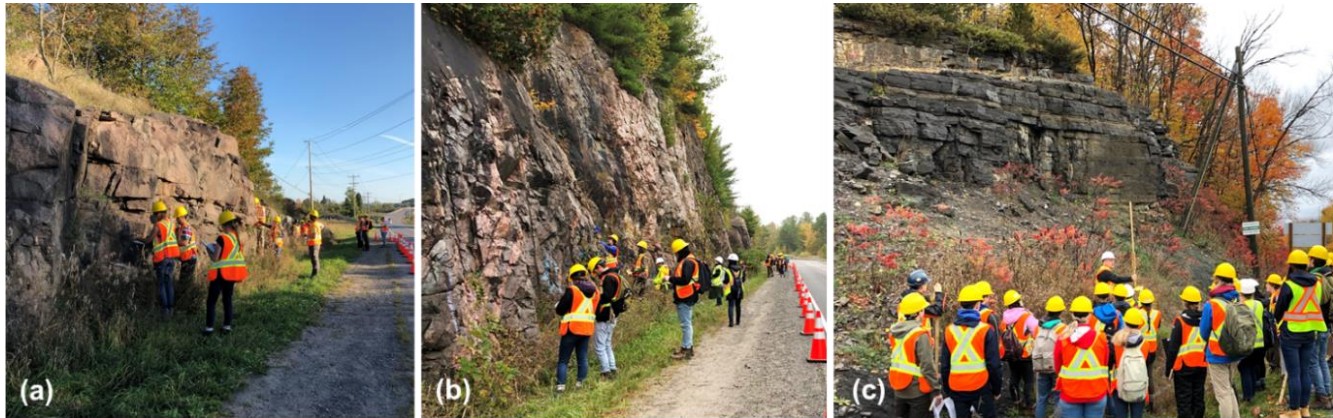

**Figure 1: Example field trip locations in the Kingston area of GEOE/L 221 in person; (a) Field Trip 2 (Joyceville), (b) Field Trip 4 (Perth Road Pluton), (c) Field Trip 6 (Moreland-Dixon Road Part 2).**

## 4. Remote Course Delivery in Fall 2020

The fall 2020 remote course delivery was implemented on an emergency basis due to the impacts of the COVID-19 pandemic. Course assessments included professionalism, lab assignments, a group project, and exams through the term (Table 3). A decision was made at the department level to reschedule fall 2020 courses from a full load of approximately 5-6 courses over 12 weeks (plus exam time) to a full load of 2-3 courses over two sub-terms of 6 weeks each (plus 1 exam week). The lecture schedule increased from 2-3 lectures per week to up to 6 lectures per week. It should be noted the recommended practice for

pre-recorded lectures, as of fall 2020, was to be up to approximately half the length of an in-person lecture. Thus, 50-minute lectures from fall 2019 became up to 25-minute lectures in fall 2020. Normally 8 lab assignments in a 12-week term were condensed to 5 lab assignments. The group project was conducted over 4 weeks instead of 6 weeks. Midterm and final exams were replaced with weekly quizzes and a final oral exam.

**Table 3: GEOE/L 221 course evaluation in fall 2020**

| Assessment Item | Time | Grade Weight |
|---|---|---|
| Professionalism, Individual | Ongoing | 5% |
| Q&A Course Engagement | Ongoing | 5% |
| Lab Assignments (5, Individual) | Weeks 1-5 | 30% (Subtotal) |
| Group Project (Written Report) | Weeks 4-7 | 30% (Subtotal) |
| Term Quizzes (6) | Weeks 1-6 | 20% (Subtotal) |
| Final Oral Exam | Week 7 | 10% |

A hybrid linear-spiral structure curriculum model of GEOE/L 221 was adopted in fall 2020 and is illustrated in Figure 2. Linear curriculum models are designed to proceed sequentially through the course in order to promote skill development and are graphically represented as a pyramid structure (Oxford Cambridge and RSA 2016; L. Anstey, pers. comm.). Spiral structure curricula are designed with a focus on central concepts and/or skills that are introduced and revisited to promote mastery as the learner moves through the course (Harden and Stamper 1999; L. Anstey, pers. comm.). The linear curriculum structure is the basis for the storyboard framework, but the internal elements of "acquisition", "practice", and "production" are rooted in the spiral curriculum structure.

In this course framework, "acquisition" components of the course include lectures, readings, and video demonstrations. As the term progressed, the number of lectures and readings reduced, and video demonstrations were emphasized in the middle of the course. "Practice" components include practice exercises that provided students with guided solutions that were not part of the course evaluation, where students had opportunities to learn and practice hands-on skills and problem-solving exercises. The "practice" components were emphasized in the first half of the course. "Production" components included lab assignments, quizzes, and group project deliverables. These occurred throughout the course and their length and complexity increased toward the end of the course. Two "major assessments" were highlighted in Week 5 (individual) and Week 7 (group) in the storyboard because they were culminating deliverables, namely the immersive virtual field mapping exercise and the desktop site investigation report on the geology of the Kingston area, respectively. The "collaboration" components occurred in the latter half of the course where students worked on the term group project. The final group report deliverable was preceded by smaller table of contents (in Week 4) and executive summary (in Week 5) deliverables.

The skills-based learning elements of the course can broadly be categorized into virtual elements, manual elements, and blended virtual-manual elements. Virtual elements included video demonstrations, manual elements included hands-on skills with compasses and drafting tools, and blended elements included use of 3D photogrammetry models and the culminating immersive virtual field mapping exercise.

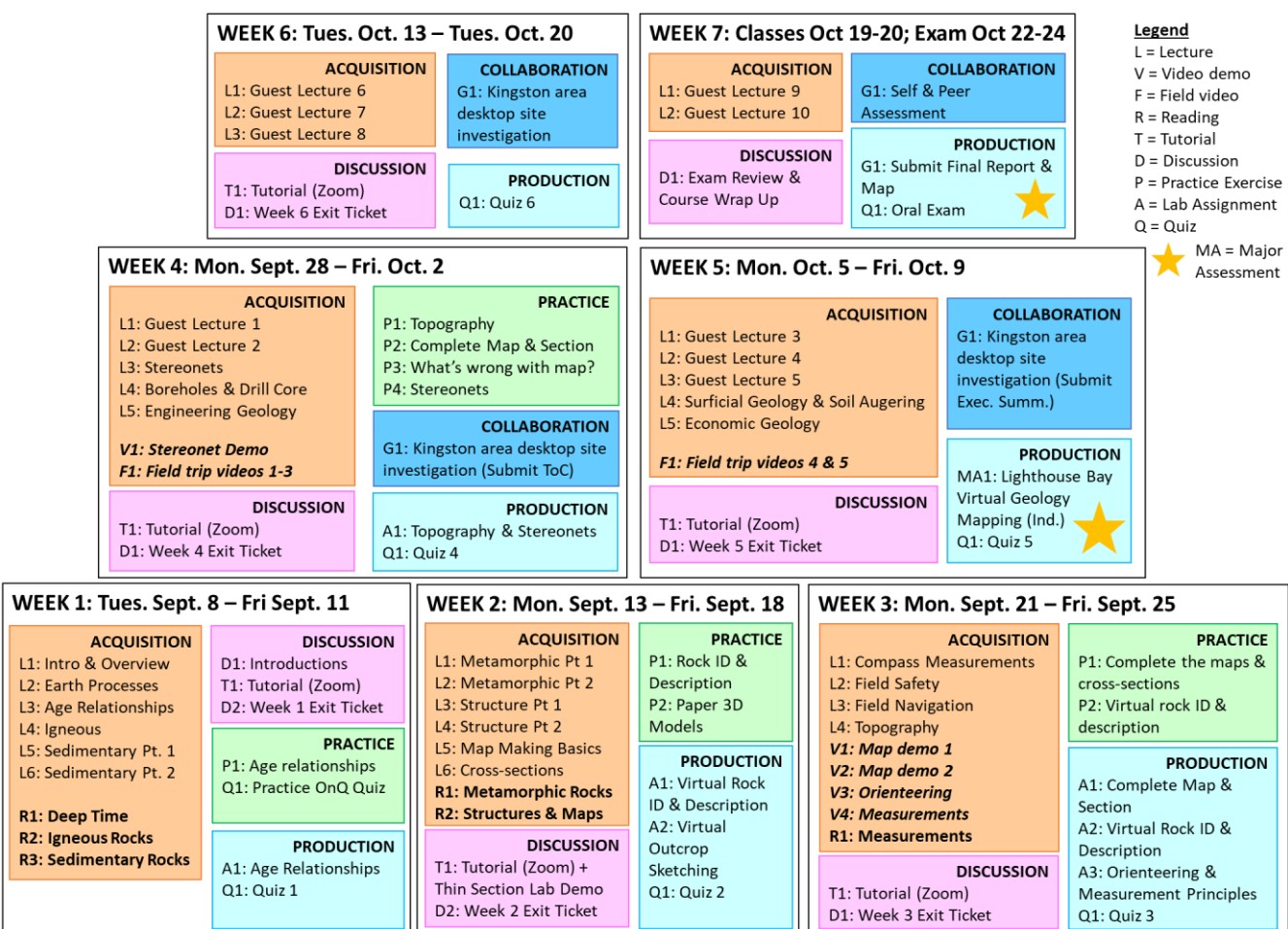

**Figure 2: Hybrid linear-spiral curriculum model storyboard of GEOE/L 221 in fall 2020 with remote delivery showing weekly course activities and deliverables.**

## 4.1 Virtual Learning Elements

The virtual learning course elements included video demonstrations and 3D photogrammetry models. Video demonstrations are an acquisition curriculum component and include field site tours, geological map and section demonstrations, and field

skill demonstrations of orienteering and structural measurements. Examples of these videos are shown in Figure 3. All videos were made available online for students to view asynchronously and as many times as they wished.

The field site videos consisted of a suite of five videos where the author introduced students to key outcrops in the Kingston area that are normally visited during the in-person version of the course (Day 2020a). In addition, the field site videos provided an opportunity to demonstrate identification of lithologies in outcrops, age relationships between geological units, and measurements of key geological structures, which supplemented other course material in the context of a real field site and in the Kingston area. Students were also able to use the field site videos as a source for their desktop study group report on the
geology of the Kingston area.

The geological map and section demonstration videos presented a real-time narrated tutorial by the author on interpreting geological models and drafting maps and sections (e.g. Day, 2020b). Two videos were created with different levels of geological difficulty. These videos provided a detailed overhead view of the steps in this exercise that rivalled the experience
of live overhead document camera demonstrations normally done during in-person tutorials, in addition to the benefit of unlimited on demand viewing opportunities.

The two field skill videos created by the author demonstrated (i) pace and compass navigation (Day, 2020c) and (ii) orientation measurements of planar and linear geological structures using two types of compasses (Day, 2020d). The user perspective of
reading measurements off a compass, aided with embedded video graphics such as arrows to direct the viewer to relevant details, provided by the videos, in addition to asynchronous and unlimited on demand availability, provided students with excellent opportunities to learn at their own pace.

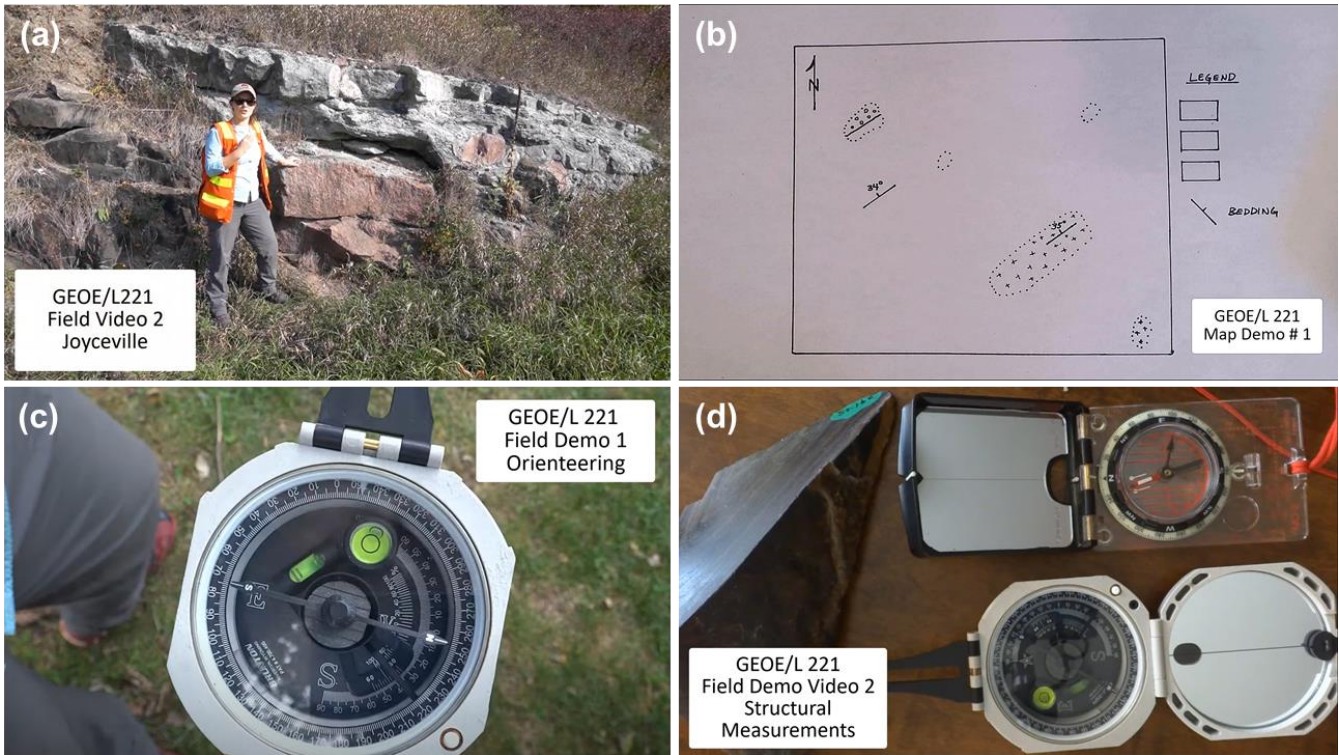

Figure 3: Examples of video recorded course content; (a) one of five field videos at a Kingston area outcrop; (b) one of three map demonstration videos; (c) field demonstration video on orienteering; (d) demonstration video on measuring orientations of planar and linear structural features (strike/dip and trend/plunge, respectively).

## 4.2 Manual Learning Elements

The manual learning course elements included pace and compass navigation, compass measurements of geological structures, hand drafted stereonet plotting and analysis, and hand drafted geological maps and sections. Students were required to purchase a geological compass, field notebook, and drafting equipment in time for the beginning of the course to use them for practice exercises and lab assignments. Purchases of this equipment are a regular cost item for both the fall 2020 remote and other in person course offerings.

### 4.2.1    Field Navigation

Pace and compass navigation skills were developed through acquisition and production curriculum components, where students learned the concept and skills through lectures and a video demonstration (acquisition) and were tasked with an independent outdoor lab assignment to conduct a closed loop traverse (production). The closed loop traverse assignment deliverable included (i) a traverse plan in students' own neighbourhoods of a 1.5-2 km route with at least 8 linear segments (in different orientations) in Google Earth (© Google, 2021), (ii) a hand drawn traverse map showing the travelled route and

205 bearing, pace count, and distance (in metres), and (iii) a graph of the students' pace factor. Examples of the first two submission items are shown in Figure 4.

Skills to measure orientations of geological structures using a compass were developed through acquisition and production curriculum components. Students learned the skills through a video demonstration, while lecture content and field site video
demonstrations discussed identification of geological features suitable for measurement during field mapping (all acquisition). Students were tasked with creating their own demonstration video for a lab assignment (production) that included measurements of both planar and linear structures, using strike/dip (right hand rule) and trend/plunge, respectively. Students measured the planar orientation of an inclined surface in their homes (strike/dip) and taped a provided paper template with a line onto the inclined surface to measure its linear orientation (trend/plunge).

Figure 4: Example lab submission of independent outdoor closed loop traverse lab; (a) Google Earth satellite image map (© Google 2021) with traverse plan and (b) hand drafted traverse map.

### 4.2.2    Hand Drafted Stereonet Data

Stereonet plotting and analysis skills were developed though acquisition, practice, and production curriculum components. Students learned the concept of stereonets through pre-recorded lectures, an instructor demonstration video (Day, 2020e), and live teaching assistant tutorial demonstrations (acquisition). Stereonet problems were included in practice exercises where solutions were provided and discussed during live tutorials (practice). Students demonstrated their use and analysis of stereonets in lab assignments and the final oral exam (production).

### 4.2.3    Hand Drafted Geological Map and Section

Students completed multiple geological map and section completion exercises through the course, with increasing difficulty each time. The first two exercises were completed by students alongside demonstration videos (e.g. Day, 2020b), one exercise during a practice activity and tutorial, and lastly the question shown in Figure 5 was part of lab assignment 3. Students were provided with guidance on structural style (of the folds), how to calculate apparent dips where needed, drawing the section in

an appropriate position and with equal vertical and horizontal scales, and drafting style (including colouring and contact line types). This assignment was marked with equal weighting of content and style (total /20) for each of the map (marked /10) and cross-section (marked /10).

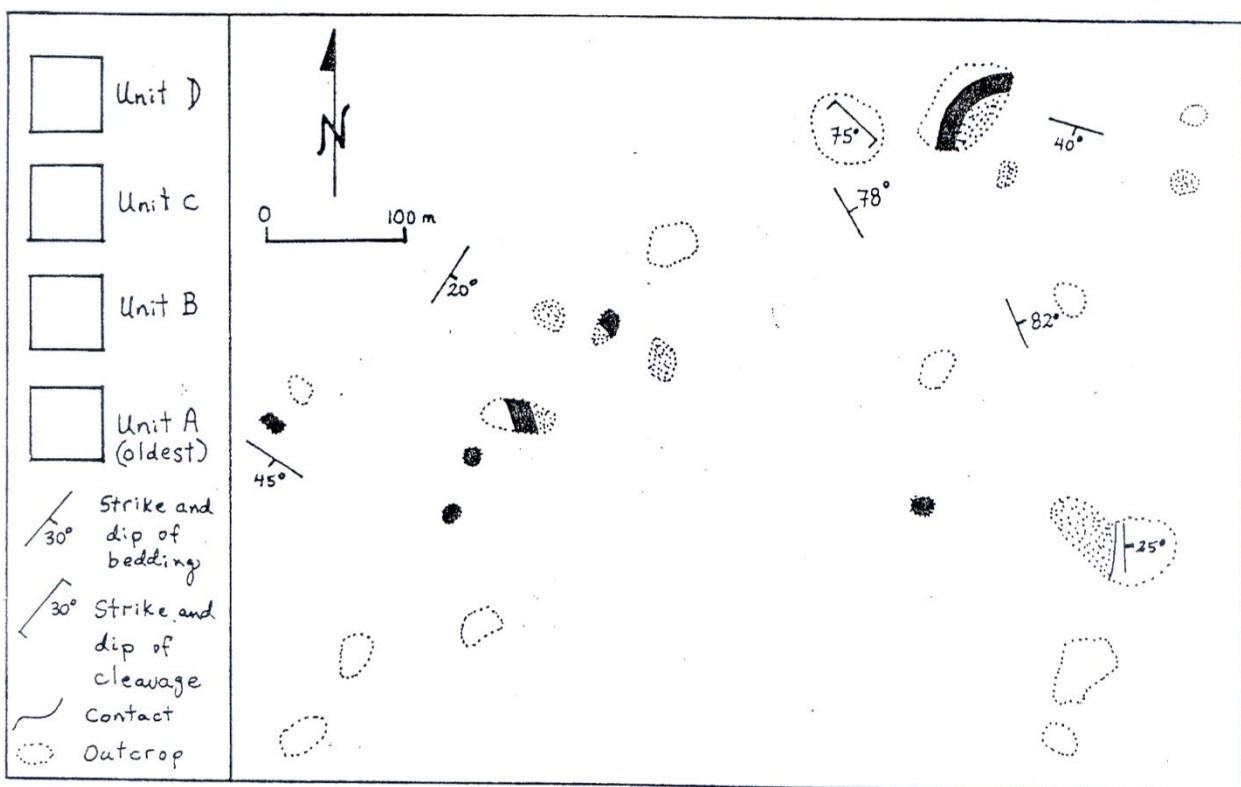

**Figure 5: Lab assignment question to complete the geological map and section.**

## 4.3 Blended Virtual-Manual Learning Elements

The blended virtual-manual learning course elements included use of virtual 3D photogrammetry models of rock hand samples and outcrops, as well as the culminating immersive virtual field mapping exercise.

### 4.3.1    Virtual Rock Samples and Outcrops

The virtual 3D photogrammetry models of rock hand samples and outcrops were used for skills development in rock observation, classification, and outcrop sketching, through acquisition, practice, and production curriculum components. Examples of hand sample 3D photogrammetry models are shown in Figure 6. Students were introduced to these skills and concepts through lectures and live teaching assistant tutorial demonstrations and discussions (acquisition). Practice opportunities were included in tutorial exercises where solutions of rock identification and classification, as well as examples

of sketches, were presented and discussed. Students demonstrated their understanding of identification, classification, and sketching of rock photogrammetry models and photographs through lab assignments, quizzes, and the final oral exam. An example of an outcrop photogrammetry model, featuring a normal fault, and an accompanying sketch submission are shown in Figure 7. Other outcrop photogrammetry models used in the fall 2020 course offering featured Jurassic sandstone with cross-

bedding from Landram Bay, East Devon Coast, United Kingdom (Mahon, 2015), and an anticline from near Lunenburg, Nova

Scotia, Canada (Young, 2020).

The virtual rock samples were selected based on what was available at the time of planning the course offering in summer 2020, and in sufficiently good quality 3D models that mineral grains or crystals could be discerned when zoomed into the sample. There was significant effort made by the Department of Geological Sciences and Geological Engineering at Queen's

University during spring and summer 2020 to create 3D photogrammetry models of many rock samples in time for fall 2020 remote teaching. As a result, many of the virtual rock samples used in GEOE/L 221 in fall 2020 were the same as those normally used in person during classes and indoor labs.

The virtual rock outcrop models were selected based on public online availability and those that showed important geological

structures that complemented the course material and field videos from the Kingston area. Cross-bedding, brittle faulting, and ductile folding are three important structural features, and the models of these features that were selected for the course show examples of these features occurring in sedimentary rocks with no to low grade metamorphism or deformation, which are not present with such high quality in the Kingston area. This exposed students to a greater variety of geological environments than what were provided in the field videos of the Kingston area.

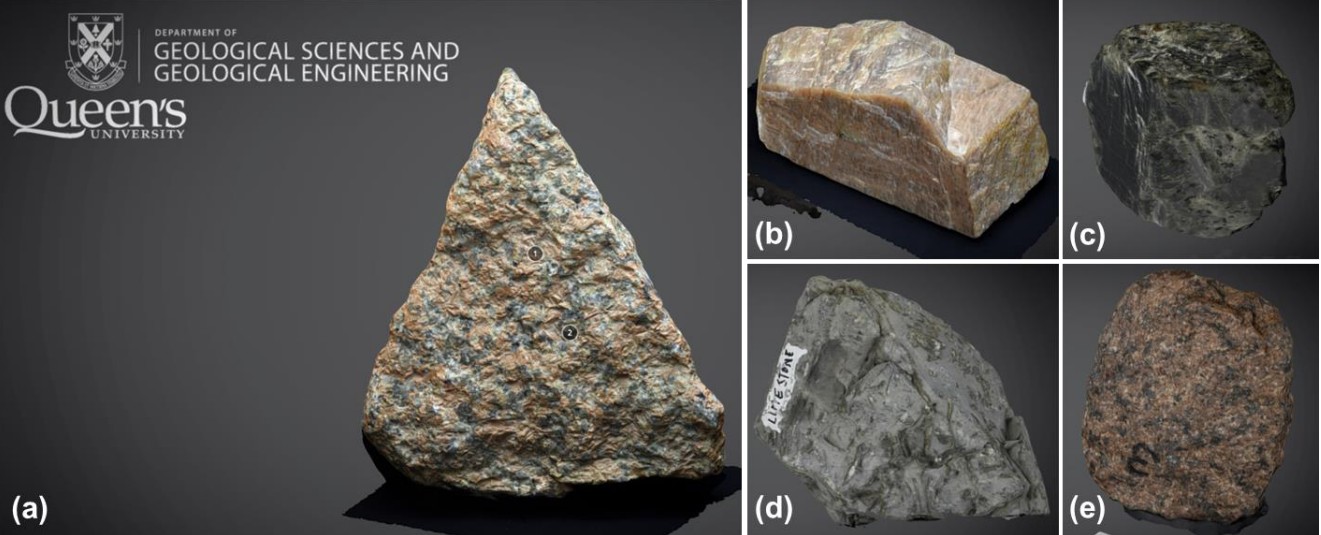

**Figure 6: Examples of hand sample 3D photogrammetry models created by the Department of Geological Sciences and Geological Engineering at Queen's University (GSGEQueens, 2020); (a) granite; (b) potassium feldspar; (c) amphibole; (d) limestone; (e) syenite.**

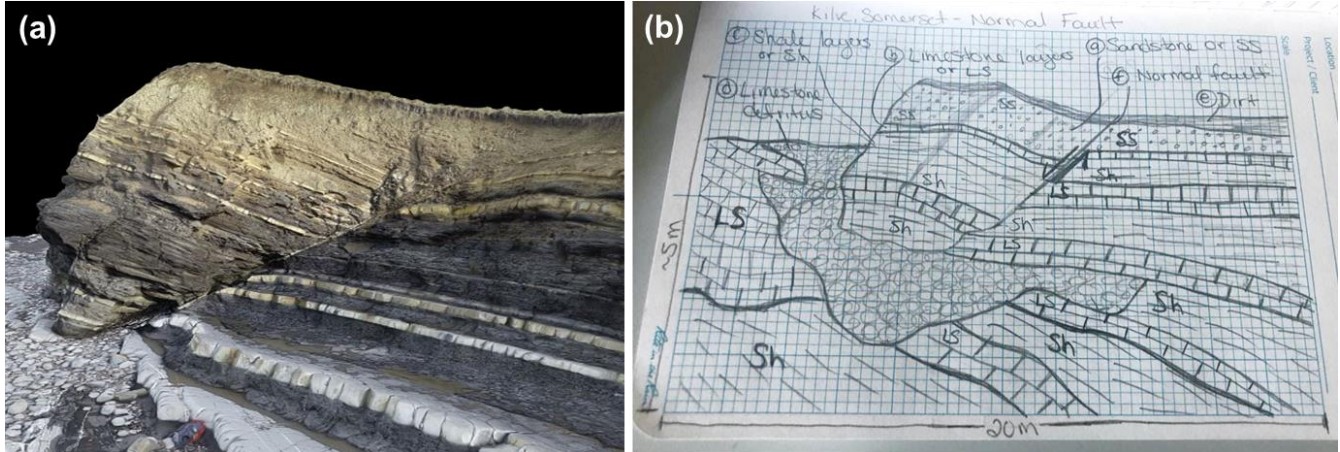

**Figure 7: (a) Example 3D photogrammetry outcrop model (Peacock, 2020) and (b) example sketch submission.**

### 4.3.2 Immersive Virtual Field Mapping

The Lighthouse Bay Virtual Landscape immersive virtual field mapping exercise by Houghton and Robinson (2017) was used as a culminating major lab assignment. The Lighthouse Bay software offers an immersive video game style experience (Figure 8a) where the user (student) is free to explore the field area with the use of embedded GPS and compass tools, discover outcrops, and collect field data that is provided by virtual field notebooks located on each outcrop. The virtual video game style experience limits users to a walking pace, so concepts of time management and traverse planning are embedded. Other field investigation aspects that are part of this experience include recording field data and sketches in personal, physical field notebooks, and using topography to guide mapping (Kingston has low topographic relief and is therefore not emphasized in the in-person field components of GEOE/L 221). Field skills that are not used or practiced, however, include manual pace and compass traversing, identification and characterization of geological features, and manual measurement of geological structures.

An accompanying base map with topographic contours and landscape features is available from Houghton and Robinson (2017) and offers the blended aspect of the learning elements. With this, students manually drafted their geological map and section interpretations based on their field investigation (Figure 8b and Figure 8c). Leading up to this lab assignment, students learned how to develop geological interpretations and complete geological maps and sections through practice exercises and earlier lab assignments.

This assignment specifically required submission of the completed geological map (marked /40 total, for content /30 and format /10), cross-section (marked /35), legend (marked /20), and calculation of true thickness for the map units (marked /5).

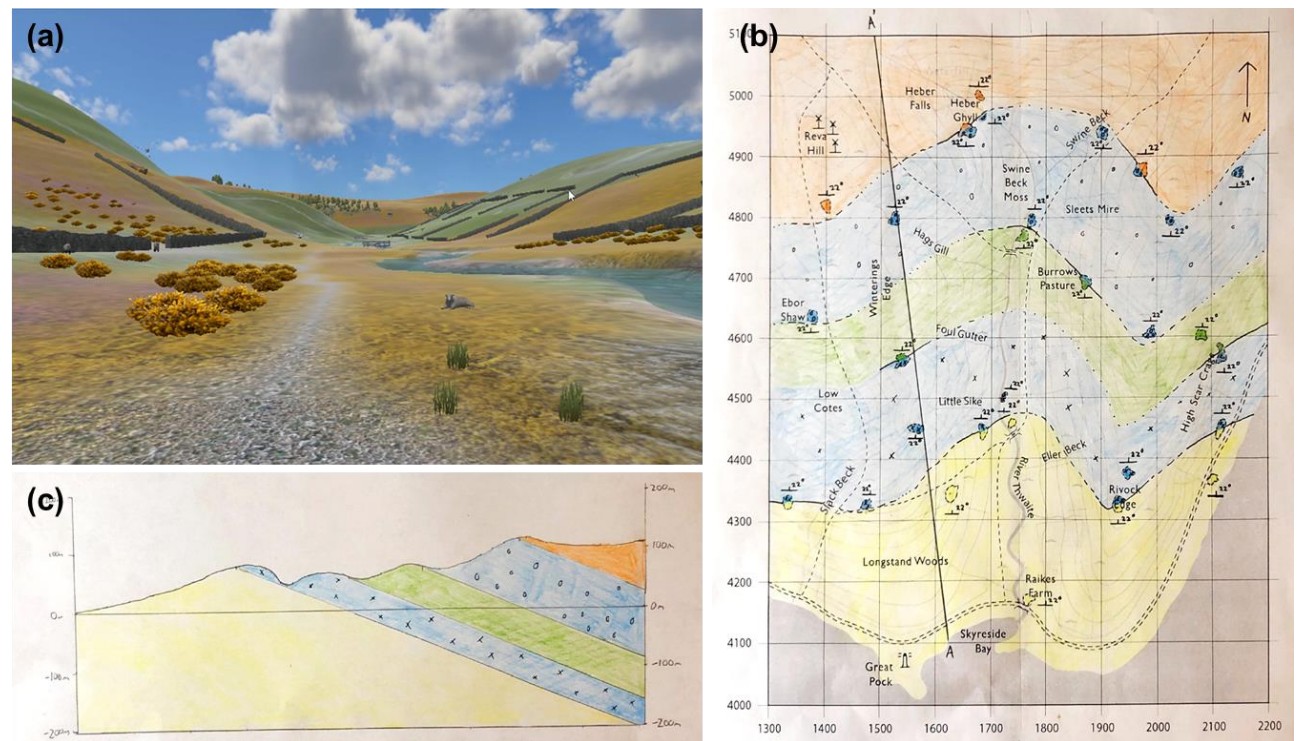

Figure 8: (a) Immersive virtual field mapping exercise, Lighthouse Bay Virtual Landscape (Houghton and Robinson, 2017); Example of a student submission of the interpreted (b) geological map and (c) cross-section.

## 4.4 Building Community

GEOE/L 221 is an important opportunity for each class of undergraduate Geological Sciences and Geological Engineering students to develop relationships and build their community that will carry them through the remainder of their degree programs. At the beginning of fall term of second year, many students in these courses have never met because they came from large common core first year programs with multiple sections for each course. Therefore, providing opportunities for students to begin building their class community was an important consideration in the remote course design for fall 2020. These included use of online, asynchronous editable documents (e.g. Google Slides and Google Docs (© Google 2021)) to facilitate a virtual gallery walk (McCafferty and Beaudry, 2017) of personal introductions, and weekly discussions about lecture and reading materials. Synchronous tutorial and lab periods were scheduled virtually by video conference calling and, for the working time during these calls, students were randomly sorted into breakout sessions of 3-4 students per group where they had opportunities to work together. In this setup, students were sorted into different groups each class with the objective being they would meet and work with all their classmates at least once during the course. Lastly, the group project was organized to have 3-4 students per group where they worked together for half of the course. A timeline of these community elements that were embedded into GEOE/L 221 in fall 2020 is illustrated in Figure 9.

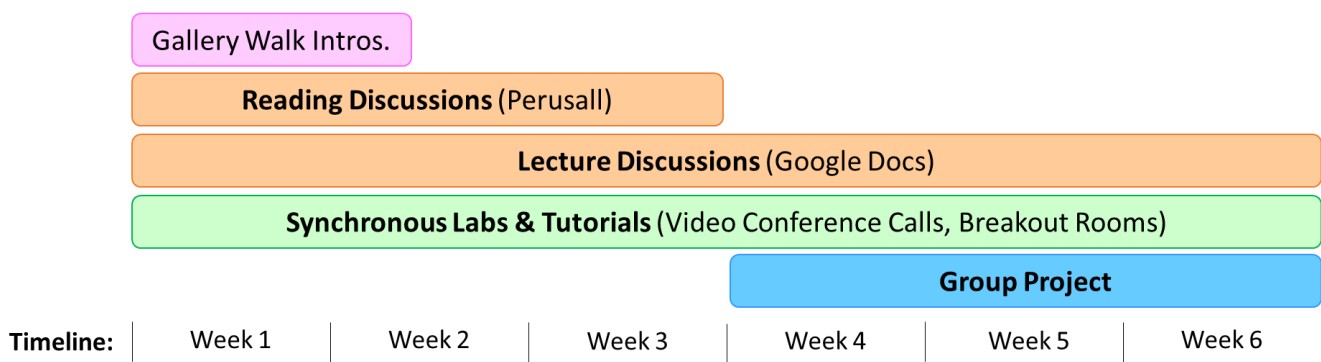

**Figure 9: Summary timeline of elements for building class community embedded within GEOE/L 221 in fall 2020.**

## 5. Analysis of Student Performance

Although there was significant change to the course deliverables and assessments in GEOE/L 221 between fall 2019, fall 2020, and fall 2021, there are a few assignments or assignment questions that remained the same, which enables comparison of student performance between these offerings and formats. The grade distributions of the geological map and section completion question (Figure 5) and the Lighthouse Bay virtual mapping assignment (Figure 8) are analyzed and discussed in this section.

The geological map and section completion question was used in fall 2019 as part of indoor lab assignment 7 (of 7), and in fall 2020 as part of lab assignment 3 (of 6). These assignments included other questions about, in 2019, stereonet data plotting and analysis of structural data, and in 2020, rock hand sample identification and an outdoor orienteering traverse. The grade distributions of this question from fall 2019 to fall 2020 show a significant drop in the mean (by 15%) but not in the maximum grade (2%), as shown in Figure 10(a). Modifying the grades so the maximum grade in each year becomes 100% does not significantly change the difference in the grade distributions between fall 2019 and fall 2020 (Figure 10(b)). These results suggest students in fall 2019 may have benefitted from their experiences during in-person field trips and accompanying assignments to measure and interpret structural data, and ultimately produce more geological maps throughout the course than students did in fall 2020. Furthermore, students in fall 2020 were learning in a more challenging remote environment and generally experiencing higher amounts of stress because of the pandemic, which may explain the 0-20% grades if students chose to focus their efforts on other tasks within the lab assignment, as this question was only worth 20 of 70 points.

The Lighthouse Bay virtual mapping assignment was used in fall 2020 as lab assignment 5 (of 6) and in fall 2021 as indoor lab assignment 9 (of 11). The grade distributions of this assignment from fall 2020 to fall 2021 exhibit an increase in both the mean (6%) and maximum grade (7%), as shown in Figure 10(c). However, when the grades are modified to set the maximum

grade in each year to 100%, the means and distributions between fall 2020 and fall 2021 become nearly identical (Figure 10(d)). These results demonstrate that in this summative assignment near the end of term in both offerings, students performed comparably, thus demonstrating equal competence in CLOs 1, 3, 4, 5, 6, and 7 near the end of the course between remote and in person learning.

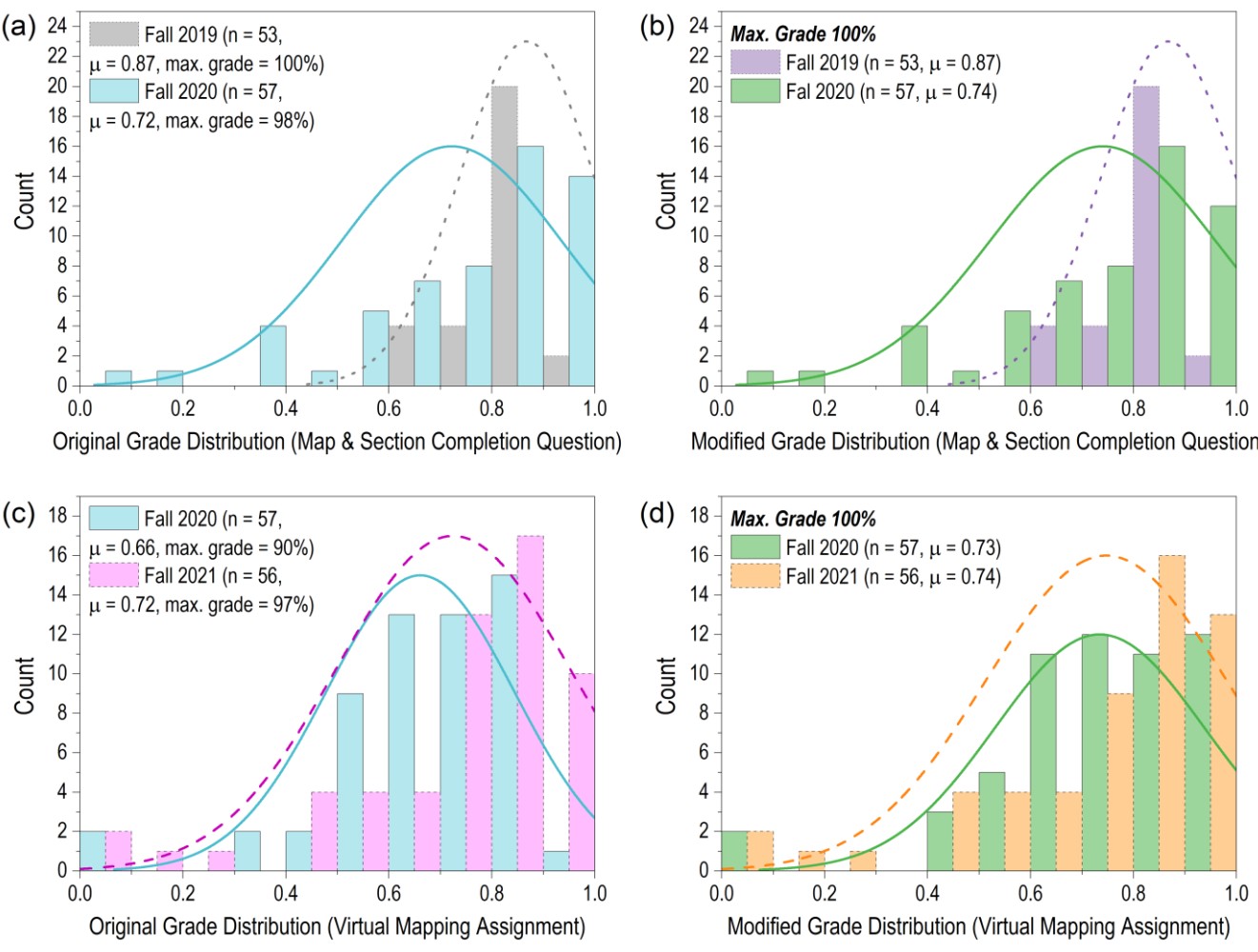

**Figure 10: Grade distributions of (a-b) a geological map and section completion question within a lab assignment in fall 2019 (in person) and fall 2020 (remote); (c-d) the Lighthouse Bay Virtual Mapping Assignment in fall 2020 (remote) and fall 2021 (in person).**

## 6. Discussion

The emergency remote offering of GEOE/L 221 Geological (Engineering) Field Methods in fall 2020 successfully achieved

coverage of all CLOs carried over from the previous in-person course offering in fall 2019. The CLOs that correspond to each assessment item are listed in Table 4. This indicates that students were successfully taught the course concepts and skills and

they were able to demonstrate their understanding in at least one course assessment. Field methods, however, is by nature a cumulative subject with a spiral structure where there is an impressive evolution of learning from basic skills for field data collection, to data analysis, integration, and synthesis into geological models and other results, and reporting the results in the form of geological maps, sections, stratigraphic columns, stereonets, and written and oral summative reports. The number of opportunities to practice various skills during the remote course offering in fall 2020 are listed in Table 5. In contrast, in-person offerings of this course provide students with "many opportunities" to learn and practice all these skills. Therefore, although the CLOs were all achieved, the missed opportunities to practice these field skills, which are normally part of the in-person course, demonstrate the need to return to in-person field methods learning.

**Table 4: CLO distribution in GEOE/L 221 assessments in fall 2020**

| Assessment Item | CLO 1 | CLO 2 | CLO 3 | CLO 4 | CLO 5 | CLO 6 | CLO 7 | CLO 8 | CLO 9 |
|---|---|---|---|---|---|---|---|---|---|
| Professionalism, Individual | X | | | | | | | | |
| Q&A Course Engagement | X | | | | | | | | |
| Lab Assignments (5, Individual) | 2 | 2 | 2 | 3 | 3 | 4 | 2 | | |
| Group Project (Written Report) | | | | | | | X | X | X |
| Term Quizzes (6) | 2 | 5 | 6 | 4 | 1 | 4 | | | |
| Final Oral Exam | | | X | X | | X | X | | |

**Table 5: Opportunities for skill development during GEOE/L 221 in fall 2020**

| Amount of Practice | Types of Skills |
|---|---|
| Many opportunities | • Rock sample observation, identification, and classification (virtual 3D model samples)<br>• Rock outcrop observation and sketching (virtual 3D model outcrops)<br>• Geological map and section completion (hand drafting)<br>• Plotting orientation data on stereonets and interpreting geological trends (hand drafting) |
| One opportunity to perform | • Pace and compass traverse navigation (outdoors)<br>• Measuring orientations of geological structures (manual compass operation)<br>• Traverse planning and time management (virtual field exercise) |
| No opportunities to learn or practice | • Rock sample observation, identification, and classification in person, with a hand lens and other field identification tools, and in the contexts of one or multiple outcrops with multiple units<br>• Identifying and measuring orientations of geological structures on outcrops<br>• Integrating pace and compass traverse navigation with geological field mapping<br>• Integrating the field data results from multiple traverses with a desktop study to report on geological and geological engineering problems |

## 6.1  Student Feedback

Overall, student feedback was positive in the context of their first fully remote term during COVID-19. Based on informal discussions and comments, students found the following remote course activities extremely helpful and valuable:

- Recorded video demonstrations combined with the live tutorials;
- Hands on practice exercises that preceded lab assignments;
- Weekly checklists that listed all course activities and deliverables;
- Asynchronous written discussion opportunities for lectures and readings; and
- The immersive virtual field lab assignment that provided a sense of what is done in the field.

Students found they learned a great deal from the course but experienced a high workload, which is partly attributed to the condensed term for this offering (7 weeks compared to a normal 12-week term). Because of this condensed term, some students also felt they did not have time to fully process and understand the material as thoroughly as they may have in a 12-week course. Other students, however, preferred the condensed term in this remote environment where they only had to focus on 2 or 3 courses at a time. Further, some students found lab assignments to be much more difficult than the practice exercises. Although the students who felt this way did not explain their reasoning, it may be because solutions to the practice exercises were explained during live synchronous video conference calls by the teaching assistants, while the lab assignments required students to complete the work independently. Certainly, the remote delivery reduces opportunities for students to approach teaching assistants or instructor with questions about assignments; however, very few students made use of the weekly office hours with the teaching assistants or instructor.

## 6.2  Delivery Experience and Recommendations

In practice, the emergency remote offering of GEOE/L 221 in fall 2020 was very challenging for both the instructor and students. The short time available to prepare the course combined with the accelerated term pace, in the context of stay-at-home orders and university closures mandated by Public Health significantly heightened the workload amount and created a stressful schedule. For example, although the weekly checklists were made to help students keep on track, it was stressful for the instructor to maintain the delivery schedule of pre-recorded lectures and frustrating for students on occasion when the lectures were not completed and posted until partway through the week.

Nearly all students engaged with all course deliverables throughout the term. In week 1, students were also very engaged with other non-deliverable course activities such as the exit tickets and adding entries to the lecture Q&A Google Doc, but this engagement significantly declined through the term, as illustrated in Figure 11. These non-deliverable course activities were partly designed to catch students who were struggling but were only partly successful as any assistance could only be provided to students who engaged in these methods of communication.

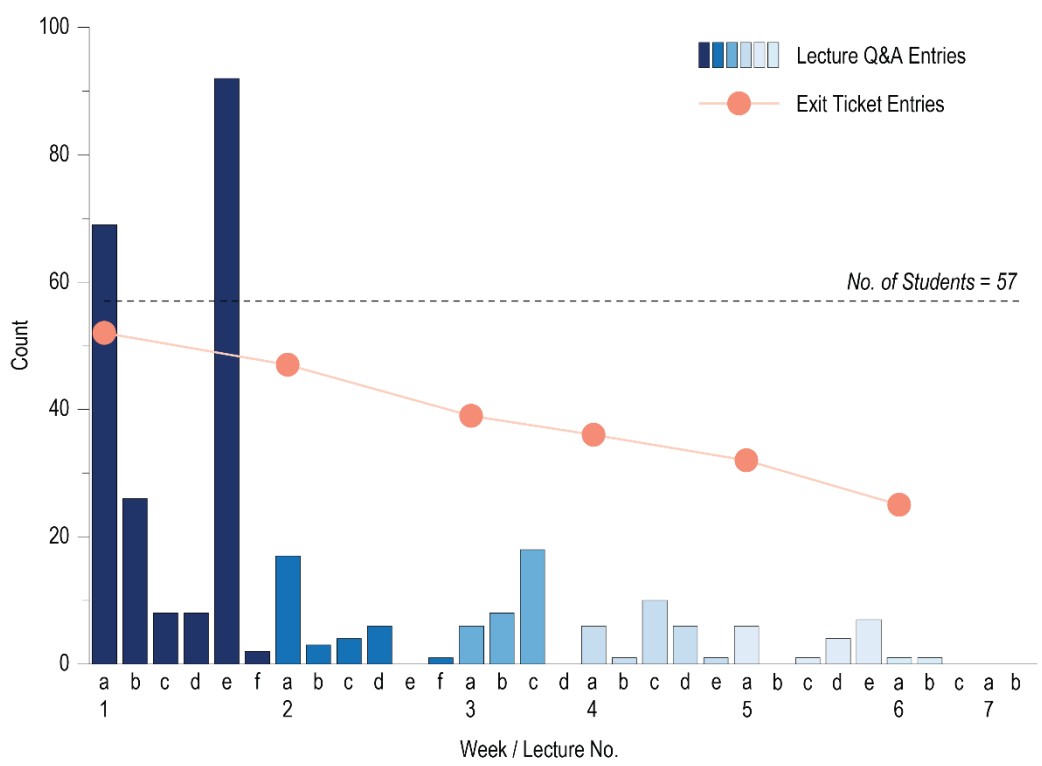

**Figure 11: Student engagement in GEOE/L 221 during fall 2020 through lecture Q&A entries and weekly exit ticket submissions.**

If GEOE/L 221 were to be delivered remotely again, the author recommends keeping many of the course elements used in fall 2020. However, live remote lectures should be used instead of using pre-recorded videos, based on the author's experience with teaching another course remotely during winter 2021. Live remote lectures were far more engaging experiences for both the instructor and students. In addition, creating pre-recorded video lectures required approximately eight times the amount of

400 preparation time when compared to preparing and delivering live lectures. Although both delivery methods require creation and delivery of the lecture material, pre-recorded video lectures also require video editing, file processing, and uploading to the online server. Students greatly appreciated the amount of contact by the instructor during GEOE/L 221 in fall 2020 to answer questions through platforms other than lectures like the lecture Q&A Google Doc, responsiveness to email inquiries, live video tutorials, and email replies to exit tickets. Whichever delivery format is used for the lectures, the author strongly

recommends providing multiple ways for students to pose questions and engage in discussions, like the ones listed above.

## 7. Conclusions

In the context of the COVID-19 pandemic, the fall 2020 emergency remote offering of GEOE/L 221 was very successful. This conclusion is primarily evidenced by the grade distribution analysis of the summative virtual field mapping lab assignment.

The course was redesigned to employ a combination of virtual, manual, and blended virtual-manual course elements, while achieving all course learning outcomes. The author (and instructor) owes the geoscience community a debt of gratitude for sharing many digital and virtual resources during the pandemic, especially those cited in this paper.

It is critical to emphasize the cumulative subject matter of field methods requires multiple opportunities to learn and practice field skills and develop an integrated understanding of related concepts. Although all course learning outcomes were achieved in this remote delivery of GEOE/L 221, many concepts and skills were learned in relatively isolated activities. The integrative aspects of learning field methods that truly require in-person field experience are lacking in this remote environment. These results demonstrate the need to return to in-person geological and engineering field methods learning as soon as it is safe to do so, in the context of the COVID-19 pandemic. It should be noted, however, that some new course elements have been integrated into in-person course deliveries since 2020 to enhance students' learning. Specifically, the weekly checklists continue to be used to help students develop their time management skills. The Lighthouse Bay virtual mapping exercise has continued to be a lab assignment since 2020 in the latter half of the term once field trips end and labs are indoors (due to weather and insufficient daylight hours). In addition, all the field and demo videos (e.g. Day, 2020a; 2020b; 2020c; 2020d; 2020e) continue to be used as supplementary material to in-person demonstrations during class and field trips. The field videos provide students with a preview of what field environments to expect and helps them plan their clothing, food, and water accordingly. The skill demonstration videos (e.g. structural measurements, map completion, stereonets) provide resources for students to review the material as many times as they wish.

To overcome the limitations of the fall 2020 remote delivery, a series of optional field trips was offered in fall 2021 for students who completed GEOE/L 221 in fall 2020, where they had an opportunity to practice and develop their integrated field skills. This activity was very well received by students and helped solidify their understanding of second year concepts and skills and add the important perspective of in-person, physical field work to their knowledge and experience for entering their third year of studies.

**Data Availability**

Access to the digital hand sample models created by the Department of Geological Sciences and Geological Engineering at Queen's University are available via Sketchfab at https://sketchfab.com/GSGEQueens.

**Author Contribution**

Jennifer Jane Day conceptualized the work, conducted the investigation, and created the written work.

## Competing Interests

The author declares that she has no conflict of interest.

## Acknowledgments

Thank you to the GEOE/L 221 students in fall 2020 for their dedication to learning in that challenging remote environment; the National Association of Geoscience Teachers (NAGT) 2020 Webinars; Teaching Assistants Gisèle Rudderham and Adriana Taylor; department colleagues Rob Harrap, Anne Sherman, Mark Diederichs, and Dan Layton-Matthews; Lauren Anstey from Queen's University Centre for Teaching and Learning; and Nerissa Mulligan, Eric Tremblay, and team from the
445 Queen's University Engineering Teaching and Learning Team.

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
