# Peer review of "Transformation of geological sciences and geological engineering field methods course to remote delivery using manual, virtual, and blended tools in fall 2020"

_Geoscience Communication, 2021_

## Author Response (AR1)

GEOLOGICAL SCIENCES
& GEOLOGICAL ENGINEERING

36 Union Street, Miller Hall/Bruce Wing
Queen's University
Kingston, Ontario, Canada K7L 3N6
Tel      613 533-2597
Fax      613 533-6592
www.queensu.ca/geol

November 21, 2022

Prof. Steven Whitmeyer
Editor
Geoscience Communication

Dear Professor Whitmeyer,

Thank you for your consideration and the detailed reviews of my submitted manuscript to the special issue on Virtual Geoscience Education Resources in Geoscience Communication. I am pleased to present to you, as enclosed, a revised manuscript and accompanying documents for the manuscript number GC-2021-18-R1, titled *"Transformation of geological sciences and geological engineering field methods course to remote delivery using manual, virtual, and blended tools in fall 2020"*.

The changes made to the original manuscript are indicated with red coloured font in the marked-up version. To address the review comments individually, the original editor and reviewer comments are provided below in italicized font and followed immediately by a response. In addition, a clean R1 manuscript that incorporates all updates is also provided.

*COMMENTS FOR THE AUTHOR:*

**Editor:**
*Comments and suggestions for this manuscript have been provided by three reviewers, all of whom view it as a potentially valuable contribution to the literature on virtual field experiences. Each of the reviewers highlights edits that could improve the manuscript, including augmenting the background rationale for the field course, expanding on the experiential aspects of the course and the lessons learned, and a more in depth discussion of assessment and rubrics used. Each of the reviewers also includes suggestions pertinent to specific sections of the manuscript, and I encourage the author to address those in the revised manuscript. After revision, I think this manuscript will likely be a valuable contribution to the SE/GC special issue on Virtual geoscience education resources.*

**Reviewer #1 (Jacqueline Houghton):**
*It is always interesting to read other people's experiences of virtual field teaching, particularly courses created quickly due to the COVID-19 restrictions. Overall, I enjoyed the paper, it is well written and goes into great detail on what was planned and the technical details of how this remote delivery field methods course was delivered. However, for others to learn from the experience, there*

*needs to be more on what actually happened in practice. I wanted to know more about the experience of delivering the course from both the instructor and student point of view. What worked, what didn't and why? For example, what difficulties did student's face using the software and how were they overcome? How well did student's engage with the activities? What checks did you have in place to catch students who were struggling? How did instructors find creating the course? What advice would you offer to others doing the same? How did student results from this year compare with previous years and can you break this down to see if there were elements where they did noticeably better or worse than when in the field? And if you had to deliver it again, if restrictions were to remain in place, what would you do differently next time?*

A new Section 6.2 and Figure 11 (Delivery Experience and Recommendations, lines 379-405) have been added to address these questions.

*You strongly recommend returning to in-person field methods teaching; something the majority of us will whole-heartedly agree with. Some virtual teaching elements can supplement and enhance in person field teaching though. They can act as pre-trip training or post-trip recaps, exercises that can supplement alternative field experiences for those unable to fully participate in fieldwork, they can even be used as "wet weather day" activities. You do mention some new course elements should be integrated into the in-person delivery, and it would be good to know which these were and why they would be a positive addition to the teaching fieldwork repertoire.*

Details about what course elements have been used during in person offerings since fall 2020 have been added to Section 7, lines 419-426.

*There are quite a few places through the text (particularly sections 3 and 5) where information could be presented as tables or figures for clarity and to reduce the need for repetition.*

Some in text information has been revised as the new Table 2 (lines 101-103) and Table 5 (lines 356-359).

*Section 3: Regular in-person course delivery. This has too much detail, given it is the virtual field course being discussed, and needs editing down. The bullet points on the field trips could be presented as a table or perhaps in an appendix if they are essential, and a similar graphic to figure 2 (which gives a lot of information in a concise manner) would condense the text and allow an easy comparison between the two.*

This section has been shortened by moving the bullet points into a new Table 2 (lines 101-103).

*Section 4: Remote course delivery. What was your thinking behind choosing these particular virtual learning course elements? Why did you choose the particular rock virtual rock samples and outcrops? Were there exercises you would have liked to have done but could not find the resources or insufficient time to create them? Out of interest – would students normally be required to purchase compass, notebook drafting equipment or was this an additional financial burden due to COVID-19?*

Comments have been added to explain why the chosen virtual rock samples and outcrops were selected, in lines 251-264. An explanation about the equipment purchases has been added in lines 197-198.

I have elected not to add discussion about what exercises I might have liked to do, had there been sufficient time or resources, as I believe it is outside the scope of the paper.

*Section 5: Discussion. There is an interesting debate to be had here; if all the learning outcomes can be covered by a remote exercise then why go in the field given the costs and accessibility issues involved? As students had no opportunities to learn or practice some techniques, it would be useful to consider whether all the outcomes were really fully achieved or whether they could only be partly achieved by remote teaching. Again, a table comparing what students would learn in the field against what they learnt in the virtual classroom would give clarity to this section. As mentioned above it would be good to have more on student feedback, for example, what was it about the lab assignments they found more difficult verses the practice exercises and is this something you see in the in-person classes too or is it unique to the remote environment?*

More details about student feedback and participation have been added to lines 374-376, 379-405, and new Figure 11. Unfortunately, there was not much student feedback beyond what I have already included in this paper. The argument for offering in-person field courses when possible instead of only virtual field learning is presented in lines 350-352, new Table 5 (lines 356-359), and lines 413-418.

*Thank you for sharing your experience of remote learning. This is an interesting paper, but it does need more on the experiential aspects of the course and the practical lessons learnt to make it a fully rounded case study. I note the comment already posted makes similar suggestions. Best wishes and fingers-crossed we can all return to the field in the near future!*

**Reviewer #2 (Eric Pyle):**

*The manuscript "Transformation of geological sciences and geological field methods course to remote delivery using manual, virtual, and blended tools in fall 2020", was submitted for review and was a delight to read, having had to do much of the same sort of work during 2020 and 2021, as well. This manuscript not only defines a contingent course design forced by public health considerations, but does so in terms of well-defined course learning objectives focused on geologic mapping skills development. Assessments are described in general and mapped to course learning outcomes. A logical spiraled course curriculum is also defined in a clear and succinct manner. Many of the course activities are traditional, transposed to an asynchronous format, while others are rather innovative and suggestive of further possible development.*

*Despite the strengths of this manuscript, there are a few shortcomings, none of which are fatal. They would simply strengthen the manuscript and broaden its utility to the field learning audience. As it stands, the manuscript is a rich narrative description of the design and delivery of what would normally be an immersive and experiential course experience, modified to fit an online delivery with asynchronous individual experiences. What is missing at the beginning is a justification of field instruction to start with, that is, providing a rationale for the considerable effort in organizing the course, rather than simply diverting students to a different type of offering.*

The requested justification has been added to Section 1, lines 55-63.

*Another shortcoming in this manuscript is the discussion of the assessments employed. There is a clear map of the assessment specifications to learning outcomes, but the specific details of the assessment instruments or tasks are limited. Scoring examinations can be straightforward, but how the artifacts or products were assessed is not presented – were there rubrics, and if so, can they be presented? An extension of the assessment discussion are the actual results, and the extent to which*

*they represent the extent to which students met the intended outcomes. Furthermore, a comparison of these results compared to prior (normal) offerings of the course would provide support to the declarative statement in Line 333, where the effort was described as "very successful." Given the relative lack of a theoretical framework or assessment results, it is difficult to accept this assertion.*

> A new analysis of student performance has been added to the manuscript (the new Section 5 and Figure 10, lines 314-342) to provide evidence to support the claim that the fall 2020 remote offering was "very successful". In this analysis, I present grade distributions for one assignment question (geological map and section completion) and one lab assignment (Lighthouse Bay Virtual Mapping Assignment) and provide explanations for the grade distributions between 2019 and 2020 (for the assignment question) and 2020 and 2021 (for the virtual mapping assignment). Information about the grading scheme of these assignments has been added to Section 4.2.3 (lines 228-235) and Section 4.3.2 (lines 291-292).

*Overall, this is a very well-written manuscript that can be made even better with a few more supporting details.*

In closing, I would like to thank the reviewers for their thoughtful comments which have certainly improved the quality of this paper. In addition to the reviewer comments above, I have reviewed the paper in full for any other spelling, grammar, referencing, or other errors. I hope the revisions presented in this revised (R1) submission are acceptable to the editor and reviewers for publication, and I look forward to your response.

Sincerely,

Dr. Jennifer J. Day, Ph.D., P.Eng., P.Geo.
1st Author & Corresponding Author

Assistant Professor
Department of Geological Sciences and
Geological Engineering, Queen's University
Kingston, ON, Canada    K7L 3C3
day.jennifer@queensu.ca

---

## Author Response (AR2)

GEOLOGICAL SCIENCES
& GEOLOGICAL ENGINEERING

36 Union Street, Miller Hall/Bruce Wing
Queen's University
Kingston, Ontario, Canada K7L 3N6
Tel      613 533-2597
Fax      613 533-6592
www.queensu.ca/geol

December 1, 2022

Prof. Steven Whitmeyer
Editor
Geoscience Communication

Dr. Solmaz Mohadjer
Executive Editor
Geoscience Communicationß

Dear Professor Whitmeyer and Dr. Mohadjer,

Thank you for your comments on my last revision of my submitted manuscript to the special issue on Virtual Geoscience Education Resources in Geoscience Communication. I am pleased to present to you, as enclosed, a further revised manuscript and accompanying documents for the manuscript number GC-2021-18-R2, titled *"Transformation of geological sciences and geological engineering field methods course to remote delivery using manual, virtual, and blended tools in fall 2020"*.

The changes made to the R1 manuscript are indicated with red coloured font in the marked-up version. To address the review comments individually, your comments are provided below in italicized font and followed immediately by a response. In addition, a clean R2 manuscript that incorporates all updates is also provided, which I hope you will find ready for publication.

*COMMENTS FOR THE AUTHOR:*

*(1) I think showing one photo as an example in Figure 1 would suffice.*
I respectfully disagree because all three photos show the diversity of geologies covered during in-person field trips in this course. I have added more details to the figure caption to emphasize the geological diversity (Line 120-121).

*(2) What is the question shown in Figure 5? Please add it to the figure caption.*
The figure caption has been edited to improve clarity (Line 236).

*(3) What are the numbers and "x" in Table 4? Please add to the caption.*
They were used to indicate single instances of CLOs. To improve clarity, the X symbols have each been replaced with the number 1 (Line 357).

*(4) Line 382, "The short time available to prepare the course..." Please insert # of weeks/months to define the short time.*
This information has been added (Line 382).

*(5) Figure 11. Replace "count" with "number of students", explain the colors (shades of blue), clarify what is week, what is lecture number.*
This figure has been revised accordingly.

*(6) Consider moving line 410-411 to the acknowledgement section.*
This has been moved to Lines 452-453.

*(7) Starting line 386, please define "engagement" and explain why you think the engagement was significantly declined through the term. This is interesting!*
I added examples of engagement in course deliverables, clarified that the analysis of engagement in Figure 11 is related to non-deliverable course activities, and added an explanation for the decline in engagement (Lines 388-394).

*(8) Line 435: "...is available..."*
Fixed (Line 442).

*(9) First line in conclusion - I suggest revising the sentence to something like "the fall 202 remote offering of GEOE/L 221 achieved coverage of all CLOs from previous in-person course offerings and yielded similar grade distribution (?) - Instead of concluding that the remote offering was "very successful" (unless you define success clearly), I suggest showing what was achieved.*
Thank you for this recommendation. I have updated the sentence accordingly (Lines 413-414).

*(10) Please include an ethics statement.*
Added in Lines 438-439.

In closing, I would like to thank the editors for their comments. I hope the revisions presented in this revised (R2) submission are acceptable for publication, and I look forward to your response.

Sincerely,

Dr. Jennifer J. Day, Ph.D., P.Eng., P.Geo.
1st Author & Corresponding Author

Assistant Professor
Department of Geological Sciences and
Geological Engineering, Queen's University
Kingston, ON, Canada    K7L 3C3
day.jennifer@queensu.ca